# Factors Affecting Implant Failure and Marginal Bone Loss of Implants Placed by Post-Graduate Students: A 1-Year Prospective Cohort Study

**DOI:** 10.3390/ma13204511

**Published:** 2020-10-12

**Authors:** Gian Maria Ragucci, Maria Giralt-Hernando, Irene Méndez-Manjón, Oriol Cantó-Navés, Federico Hernández-Alfaro

**Affiliations:** 1Department of Oral and Maxillofacial Surgery, Universitat Internacional de Catalunya (UIC), 08017 Barcelona, Spain; mariagiralth@gmail.com (M.G.-H.); manjon.irene@gmail.com (I.M.-M.); h.alfaro@uic.es (F.H.-A.); 2Department of Restorative Dentistry, Universitat Internacional de Catalunya (UIC), 08017 Barcelona, Spain; oriolcanto@uic.es; 3Institute of Maxillofacial Surgery, Teknon Medical Center, 08022 Barcelona, Spain

**Keywords:** marginal bone loss, dental implant, keratinized tissue

## Abstract

Statement of the problem: Most of the clinical documentation of implant success and survival published in the literature have been issued by either experienced teams from university settings involving strict patient selection criteria or from seasoned private practitioners. By contrast, studies focusing on implants placed and rehabilitated by inexperienced post-graduate students are scarce. Purpose: To record failure rates and identify the contributing factors to implant failure and marginal bone loss (MBL) of implants placed and rehabilitated by inexperienced post-graduate students at the one-year follow-up. Material and Methods: A prospective cohort study was conducted on study participants scheduled for implant therapy at the International University of Catalonia. An experienced mentor determined the treatment plan in accordance with the need of each participant who signed an informed consent. All surgeries and prosthetic rehabilitation were performed by the post-graduate students. Implant failure rate, contributors to implant failure, and MBL were investigated among 24 variables related to patient health, local site, and implant and prosthetic characteristics. The risk of implant failure was analyzed with a simple binary logistic regression model with generalized equation equations (GEE) models, obtaining unadjusted odds ratios (OR). The relationship between MBL and the other independent variables was studied by simple linear regression estimated with GEE models and the Wald chi^2^ test. Results: One hundred and thirty dental implants have been placed and rehabilitated by post-graduate students. Five implants failed before loading and none after restoration delivery; survival and success rates were 96.15% and 94.62%, respectively. None of the investigated variables significantly affected the implant survival rate. At the one-year follow-up, the mean (SD) MBL was 0.53 (0.39) mm. The following independent variables significantly affected the MBL: Diabetes, implant depth placement. The width of keratinized tissue (KT) and probing depth (PD) above 3 mm were found to be good indicators of MBL, with each additional mm of probing depth resulting in 0.11 mm more MBL. Conclusion: The survival and success rates of dental implants placed and rehabilitated by inexperienced post-graduate students at the one-year follow-up were high. No contributing factor was identified regarding implant failure. However, several factors significantly affected MBL: Diabetes, implant depth placement, PD, and width of KT. Clinical Implications: Survival and success rates of dental implants placed and rehabilitated by inexperienced post-graduate students were high at the one-year follow-up, similar to experienced practitioners. No contributing factors were identified regarding implant failure; however, several factors significantly affected MBL: Diabetes, implant depth placement, PD, and KM.

## 1. Introduction

Dental implants to replace missing teeth have become a predictable treatment modality for partially and totally edentulous patients; a long-term survival rate of 95.2% has been documented [1]. In contrast to implant survival, implant success has been defined in relationship to the amount of marginal bone loss (MBL) occurring over time [2]. Several etiological factors affecting MBL have been described in the literature, which include, among others: Amount of keratinized tissue (KT), gingival thickness, prosthetic abutment height, plaque accumulation, and occlusal overload [2,3]. Smoking habits and patients with a previous history of periodontal disease have also demonstrated more susceptibility to peri-implantitis [4,5]. Some strategies have been proposed to reduce or stabilize the MBL, e.g., the use of the platform-switching feature at the implant-abutment junction [6,7,8]; the use of prosthetic abutments, as a titanium base or multiunit abutments of >2 mm [9,10,11]; and achieving a mucosa thickness >2 mm at implant placement [12,13]. 

The vast majority of the literature documenting survival rates, success rates, and MBL of implant treatment has been published by experienced teams, in university settings with strict selection criteria or in private offices [1,2,3,4,5,6,7,8]. Presently, several million implants are placed each year in patients worldwide; most of them are inserted by practitioners for whom implant treatment is not a daily activity [14]. The literature issued on dental implants provides a picture of the optimal performances that ought to be achieved when implants are placed by well-trained and -skilled clinicians [1]. This might not be representative of the status of implant therapy when performed in private clinical practice in the hand of less experienced clinicians. In a retrospective study comparing experienced and nonexperienced surgeons, Preiskel and Tsolka showed that experience had a major impact on the probability of implant failure [15]. More recently, Sendyk et al. [16] concluded that implant failure was significantly affected by the experience gained by the surgeon and the number of implants placed, less or more than 50 implants. In a foreseeable fashion for immediate loading protocols, Ji et al. [17] found an increased risk of implant loss in the hands of surgeons with less than five years of experience, and 12.2% compared to 2.4% in the hands of more experienced surgeons.

Therefore, the purpose of this prospective study was to evaluate the success, survival, and marginal bone loss of implants (C1, MIS Implants Technologies, Shlomi, Israel) placed and rehabilitated by inexperienced post-graduate students without applying strict selection criteria. In addition, other variables were investigated and correlated to patient health, local site, and both the surgical and prosthetic protocols.

## 2. Materials and Methods

### 2.1. Study Design

This prospective cohort study was conducted at the Department of Oral and Maxillofacial Surgery of the Universitat Internacional de Catalunya after approval by the Ethics Committee of the university (CIR-ECL-2015-06). It enrolled study participants in need of implant therapy who received C1 implants from January 2016 to January 2017.

### 2.2. Inclusion/Exclusion Criteria

All indications, single crown, fixed partial dentures, and complete-arch rehabilitation, were covered. Prior to participation, the purpose and procedures were explained in detail.

Inclusion criteria were the following:

(a) Patients older than 18 years in need of implant therapy, (b) good general/systemic health (ASA type I, II), (c) patients who committed to attend all visits of the study, (d) who underwent or required a bone regeneration procedure, horizontal or vertical guided bone regeneration with or without resorbable membrane or block graft, (e) sinus lift, (f) adequate oral hygiene with FMPS (full mouth plaque score) <15% before surgery, (g) absence of uncontrolled periodontal disease, (h) agreeing to sign an informed consent.

Exclusion criteria were the following:

(a) patients with a contributing medical history in which any surgery, disease, condition, or medication might compromise the soft and hard tissues healing (noncontrolled diabetes, liver function disorder, immune system disease, (b) immunosuppressant drugs, (c) toxic habits other than smoking that might compromise or affect healing, (d) patients who have undergone chemotherapy or radiation treatment during the previous 5 years comprising the head and neck area, (e) corticosteroids therapy or any other medication that could influence postoperative healing and/or osseointegration, (f) bisphosphonate or Denosumab therapy (Prolia^®^, Amgen Europe B.V. Breda(NL) Netherlands), (g) inability or unwillingness to attend follow-up visits, (h) patients unwilling to sign an informed consent form.

### 2.3. Surgical Procedure

All the surgical and prosthetic procedures were performed at the Clinic of Dentistry of the University by 24 post-graduate students that had just completed the dental school curriculum; all were 24 to 27 years old. Before implant placement, the diagnostic protocol included a diagnostic wax-up in order to obtain a radiological guide. A cone-beam computed tomography (iCAT^®^, Imaging Science International, Hatfield, PA, USA) scan was taken in the target area with the respective radiographic guide to obtain a 3D image and for implant selection and 3D positioning. The drilling sequence was performed using each drill including the final drill that is delivered with each implant, according to the recommendation of the manufacturer. The main C1 implant characteristic is a conical shape geometry, micro-rings at the neck, and a dual thread design; the presence of platform switching and a conical 12° connection. C1 implants surface treatment is sand-blasted and acid-etched. Depending on the insertion torque recorded with the surgical motor (Implantmed W&H, Burmoos (AU) Austria), greater or less than 35 N cm, either a healing abutment or a cover screw was placed. The 1-stage protocol was performed by placing a healing abutment and tissue approximation using single stitches; for the 2-stage protocol, a cover screw was placed and primary flap closure was achieved over it. Patients received antibiotic (875/125 mg of Amoxicillin/Clavulanic acid, 3x/d during 7 days; in the case of penicillin allergy, 300 mg of Clindamycin every 6 h during 7 days) and analgesic anti-inflammatory treatment (600 mg Ibuprofen 3x/d); rinsing with Chlorhexidine (0.12%) (Dentaid. PerioAid 0.12%) was prescribed 2x/d for 2 weeks.

After 7 days, the patients were recalled for suture removal and then again after one month. After 3 months of healing in the mandible and in the maxilla, osseointegration was checked clinically and radiographically. The prosthetic phase (T1) was started and fixed partial ceramo-metallic prostheses seated on multi-unit abutment and single ceramo-metallic crowns bounded to a titanium base were prepared.

### 2.4. Study Variables and Measurements

Demographic parameters of the participants such as age, sex, smoking status, and medical history were recorded.

Twenty-four variables were recorded according to the characteristics of the participant, the implant, the surgical site, and the prosthetic outcomes/variables (Table 1). Success rates were calculated according to the criteria of Buser et al. [18] and modified by Albrektsson and Zarb [19].

Periapical radiographs of each implant were acquired with an intraoral dental film using a plastic index according to the parallel technique immediately after implant placement, at prosthesis delivery, and at the 1-year follow-up. Measurements were performed with the Image J software (10.8.0_172, NIH, MD, USA); internal calibration was provided by the implant diameter at the neck level. At each time point, the distance from the implant shoulder to the first bone-implant contact was measured on the mesial and distal sites. The difference between baseline and the milestone served to calculate the MBL on each side. Subsequently, the mean value of the two measurements was calculated for each implant. (Figure 1).

Peri-implant clinical parameters were assessed at three sites (mesial, buccal, and distal) with the use of a periodontal probe (UNC 15, Hu-Friedy):-Probing depth (PD) in millimeters was measured from the peri-implant mucosal margin to the bottom of the peri-implant sulcus,-Bleeding on probing (BoP) was determined as presence or absence of bleeding 15 s after gentle probing,-Keratinized tissue (KT) width in millimeters was measured with a periodontal probe at the mid-buccal aspect of the implant from the free gingival margin to the muco-gingival junction. Furthermore, the KT measurements were categorized in two groups, group 1 when KT ≥ 2 mm and group 2 when KT < 2 mm.-Depth of implant placement: On the day of the surgery, a periapical radiograph was performed to provide the depth of implant placement read on the proximal sides.

In addition, implant location (maxilla or mandible, anterior or posterior), and the type of the implant-supported fixed dental prosthesis single crown (SC) or fixed partial denture (FPD) were also recorded.

Two investigators (G.M.R. and M.G.-H.) independently evaluated the clinical parameters at the 1-year follow-up. If any differences were aroused, the scores were then discussed with a third person (F.H.-A.). Clinical and radiographic examinations were performed following the same procedures at baseline (T0), at prosthesis delivery (T1), and at the 1-year follow-up (T2).

### 2.5. Statistical Methods

Descriptive data of the parameters analyzed at participant and implant levels were: Mean (standard deviation), minimum, maximum, and median for the continuous variables, absolute frequencies and percentages for the categorical ones. The probability of failure at the implant level based on each of the independent factors and covariates was determined with a simple binary logistic regression with generalized equation equations (GEE) models obtaining unadjusted odds ratios (OR) as a function of the factors of profile.

The relationship between MBL and the independent variables was investigated using a simple linear regression estimated with GEE models and the Wald chi-square statistical test. The variables were categorized as significant (*p* < 0.05) or relevant (*p* < 0.1) and a multiple-model was proposed to obtain fully adjusted coefficients.

## 3. Results

First, 130 implants were placed in 67 participants (43 women and 24 men) with mean age 48.6 (10.2) years; 12.1% were smokers with less than 10 cigarettes/day; 39.4% had a previous history of periodontal disease that was under control at the time of implant treatment; 6.1% suffered diabetes. Thirty-seven implants (28.5%) were placed in the anterior zone and 93 (71.5%) in the posterior zone; 44.6% were placed in the maxilla.

### 3.1. Implant Survival

Before loading, five implants failed in five participants, two women (3.9%) and three men (3.8%); none afterwards. Implant survival rate was 96.15% with 95% CI (91.3–98.7%); implant failure concerned 7.5% of the participants, 5% of the smokers and 2.8% of the nonsmokers. Failure at participants with or without a previous history of periodontal was 4.6% and 1.6%, respectively (Table 2).

No parameter was found to significantly affect the survival rate (Table 2). However, a relevant association between survival and implant position (anterior vs. posterior) was found; the failure risk of implants located in the posterior area was four times lower than that in the anterior one (*p* = 0.143) (Table 2).

### 3.2. Marginal Bone Loss and Implant Success

MBL was calculated on the mesial and distal sides (Figure 1); it was then averaged for each implant. The mean MBL on the mesial and distal sides was 0.48 (0.42) and 0.56 (0.43) mm, respectively; the averaged MBL of both sides was 0.53 (0.39) mm (Figure 2). The success rate was 94.62%.

MBL was affected in a statistically significant way by diabetes, 0.55 (0.40) vs. 0.32 (0.27) mm, *p* = 0.035, for patients without diabetes vs. for patients with controlled diabetes, respectively. Deeper implant placements when measured on the proximal sides led to an increase in bone loss (*p* = 0.003) (Figure 3); every mm of deeper placement increased MBL by 0.28 mm.

MBL was affected by the length of KT measured at the one-year follow-up (*p* = 0.004). Sites with KT < 2 mm led to more bone loss than sites with KT ≥ 2 mm (0.78 (0.40) vs. 0.45 (0.36) mm, *p* = 0.001, (Table 3, Figure 4); every mm less than 4 mm of KT led to an increased MBL of 0.08 mm.

Increased MBL was associated with a PD deeper than 4 mm (*p* = 0.008); each additional mm of PD above 4 mm led to an increased MBL of 0.18 mm (Figure 5).

Relevant parameters affecting MBL but not in a statistically significant way were implant diameter (*p* = 0.064) and jaw (*p* = 0.078). Implants with a larger diameter displayed less MBL than smaller ones (Figure 6): 0.73 (0.04) mm for Ø 3.3 mm, 0.59 (0.44) mm for Ø 3.75 mm, 0.49 (0.37) mm for Ø 4.2 mm, 0.38 (0.36) mm for Ø 5 mm. Implants placed in the maxilla displayed more MBL than in the mandible, 0.60 (0.46) vs. 0.40 (0.33) mm, respectively.

Finally, history of periodontal disease (*p* = 0.348), smoking (*p* = 0.617), mucosa thickness (*p* = 0.384), gingival phenotype (*p* = 0.462), insertion torque (*p* = 0.344), abutment height (*p* = 0.146), and all prosthetic variables (*p* = 0.952) did not affect the MBL (Table 3 and Table 4).

## 4. Discussion

Only a few studies have addressed the relationship between experience of the surgeon and implant survival rates [16]. These authors found that being trained as a specialist did not provide an indication of clinical achievement; rather, experience with at least 50 placed implants showed to be more determinant in this regard. Although it seems intuitive to correlate between experience and failures, many other confounding parameters, like skills of the surgeon, user-friendly implant design, and adapted drilling tools, may also affect the survival rate [17]. This is the reason why extrapolating clinical achievements from one implant system to another may be hazardous. This prospective cohort study was performed to evaluate the survival and success rates of the C1 (MIS) implant system placed and rehabilitated by inexperienced post-graduate students; the implant design is self-tapping tapered with a conical connection and platform-switching. At the one-year follow-up, the survival and success rates were 96.15% and 94.62%, respectively. These data are similar to survival features reported otherwise in the literature [20,21,22]. More specifically, they compare well to a one-year study [23] dealing with 117 C1 implants placed in 60 patients by 7 experienced periodontists; the reported survival and success rates of these active private practice practitioners were 98.3% and 95.4% respectively. Thus, this study with 24 distinct students having placed a total of 130 implants shows that the lack of experience of the surgeons did not lead to a higher implant failure rate with this implant. It may be that the presence of an experienced supervisor who helped with implant planning can explain the positive outcome; however, this would then rely on the assumption that failures are the result of insufficient planning rather than purely surgical skills. It is also possible that the C1 implant design is user-friendly in terms of drilling and achieving primary stability and is error-forgiving for the beginner.

No co-variate significantly influenced the survival rate, but a tendency was found for implants located in the posterior area to fail four times less than in the anterior one (*p* = 0.143).

The mean MBL of the implants placed by the students was 0.48 (0.42) and 0.56 (0.43) mm on the mesial and distal sides, respectively. A similar MBL of 0.54 (0.55) and 0.52 (0.45) mm was measured at the mesial and distal sites of the same implant, respectively, when placed by experienced surgeons [23]. Only two parameters, diabetes and implant depth placement, significantly affected MBL. Influence of diabetes on MBL was assessed in a recent systematic review by Maior et al. [24]; the present data are in line with these authors that reported a positive effect of the disease on MBL when it is controlled. It should be stressed that the inclusion criteria of this study allowed to enroll only controlled diabetic patients; therefore, no conclusion for patients with uncontrolled diabetes can be drawn.

Subcrestal implant placement instead of crestal has been suggested in order to avoid or delay exposure with time of the rough surface of the implant neck to the oral cavity [25]. This strategy can be successful provided that MBL is limited and that, with time, bone still covers the rough neck surface and protects it from bacterial contamination. The present data showed that implants placed deeper led to an increased MBL. Accordingly, implants placed 1.5 mm subcrestally lost an average of 0.86 (0.61) mm after one year, but this still provided an average height of 0.64 mm of bone protection over the implant neck. The implants placed crestally lost 0.31 (0.29) mm on average, but this would mean that the implant collar was no more covered with bone. If this feature remains stable with time, subcrestal placement should be advantageous from a clinical perspective.

Previous studies compared subcrestal to crestal implant placement; Ercoli et al. [26] measured MBL after a period of 12–18 months. Subcrestally placed implants lost more bone than those crestally placed, but the difference was not statistically significant. In line with our finding, they concluded that subcrestal position of the implant at the time of surgery leads to reduced odds of having implant threads exposed [26]. In a similar way, a recent randomized clinical trial assessed the influence of apico-coronal implant position after three years of follow-up [27]. No visible clinical difference between the crestal and subcrestal position was observed in terms of pocket depth; however, the radiologic parameters between the crestal and subcrestal position were dissimilar; 53.4% of the crestally placed implants lost an average of 0.29 (0.35) mm of bone beyond the implant neck. The subcrestal group lost less bone beyond the collar, 0.09 (0.18) mm on average, and frequency of this feature was lower by 25.8%, i.e., approximately half compared to the crestal implants. In the subcrestal group, 74.2% of the implants still had 0.89 (0.37) mm of bone over the implant neck; the authors concluded that subcrestal implants reduced rough surface exposure by 88% when compared to the crestal implants. What is noteworthy is that no clear relationship between exposed rough surface and bleeding upon probing was identified, although a certain statistical trend (*p* = 0.135) was identified [27]. On the other hand, a systematic review by Valle et al. [28] found significantly less MBL for implants placed subcrestally, by 0.18 mm only. The limitations of the present and also other studies are that all these bone data focus on the proximal sides; they do not address the events occurring at the buccal bone lamella, which is critical for the support and preservation of the marginal gingiva over time.

The importance of KT in the maintenance of peri-implant tissues health and MBL has been a matter of controversy. Several studies have shown higher plaque scores and more peri-implant tissue inflammation at sites with KT < 2 mm [29,30,31]. In addition, some authors suggested that the presence of a band of KT ≥ 2 mm is necessary to prevent the progression of MBL and maintain peri-implant health over time [32,33,34]. On the other hand, review papers produced limited evidence in support of the need for wide bands of nonmobile keratinized tissues around implants to maintain health and tissue stability [35,36]. Our data suggest that after one year, bone loss might be slightly more pronounced at sites with KT < 2 mm when compared with sites with KT > 2 mm, 0.78 (0.40) vs. 0.45 (0.36) mm (*p* < 0.001), respectively. A similar tendency was found by Pellicer-Chover et al. [25] for subcrestally placed implants after three years of follow-up.

Other variables established in the literature to affect MBL as mucosa thickness and abutment height [9,10,11,12,13,37,38,39,40,41] were not found to be contributing in this pool of patients and sites. What is noteworthy is that the effect of mucosal thickness and abutment height on MBL is not systematically assessed, and the reason for these discrepancies is not fully understood [37,42,43].

## 5. Conclusions

Within the limitations of this short-term cohort study, the following conclusions may be drawn:The failure rate of C1 implants placed and rehabilitated by inexperienced students at the one- year follow-up was low (3.6%) and comparable to the data obtained by experienced practitioners.No contributing factors specific to the inexperience of the students could be identified regarding implant failure and MBL.Several factors have been shown to affect MBL such as diabetes, implant depth, PD, and KT. By contrast, thickness of the gingiva and prosthetic abutment height were not found to be contributing factors.

## Figures and Tables

**Figure 1 materials-13-04511-f001:**
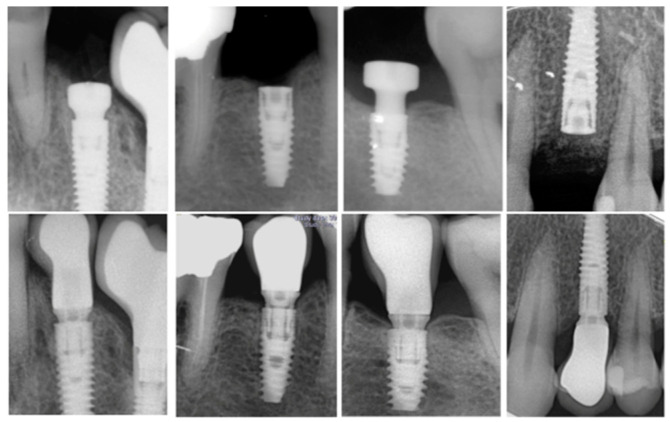
Periapical radiographs with an intraoral dental film using a plastic index according to the parallel technique for marginal bone loss (MBL) analysis between baseline and 1 year follow-up.

**Figure 2 materials-13-04511-f002:**
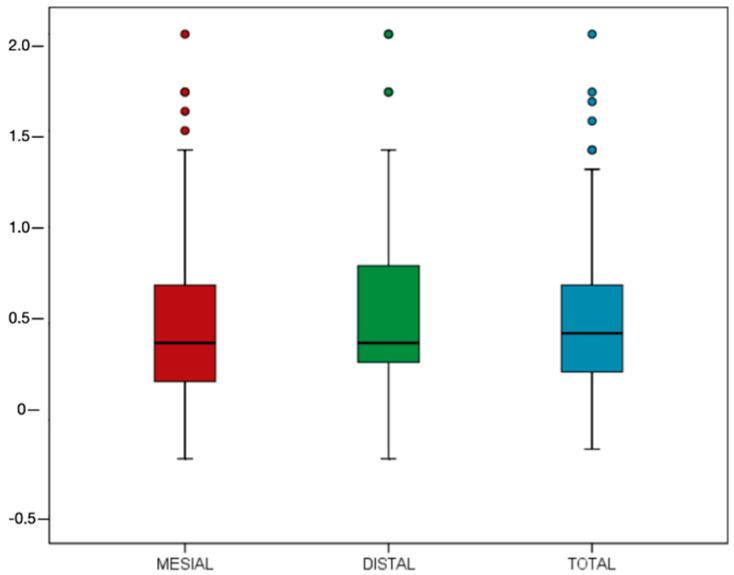
Marginal bone loss measured on the mesial and distal sides and mean of both sides.

**Figure 3 materials-13-04511-f003:**
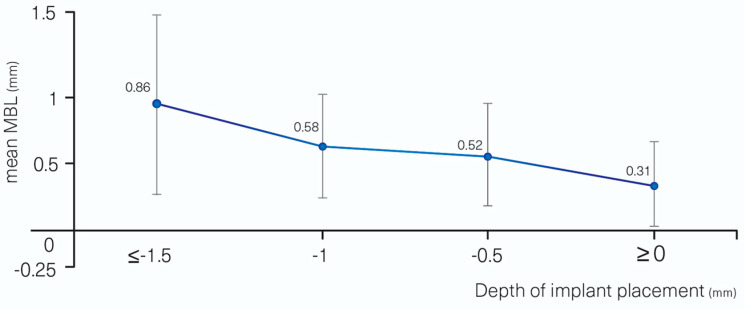
MBL vs. depth of implant placement.

**Figure 4 materials-13-04511-f004:**
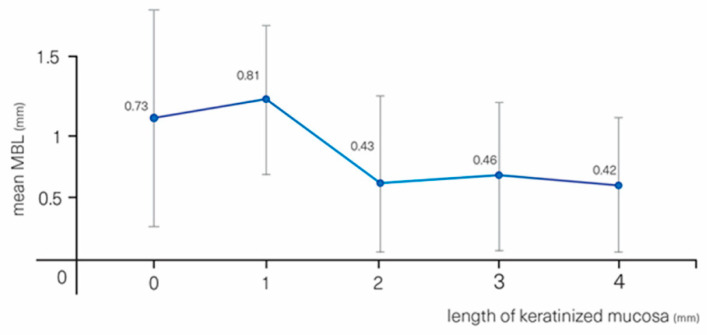
MBL vs. measured keratinized tissue length.

**Figure 5 materials-13-04511-f005:**
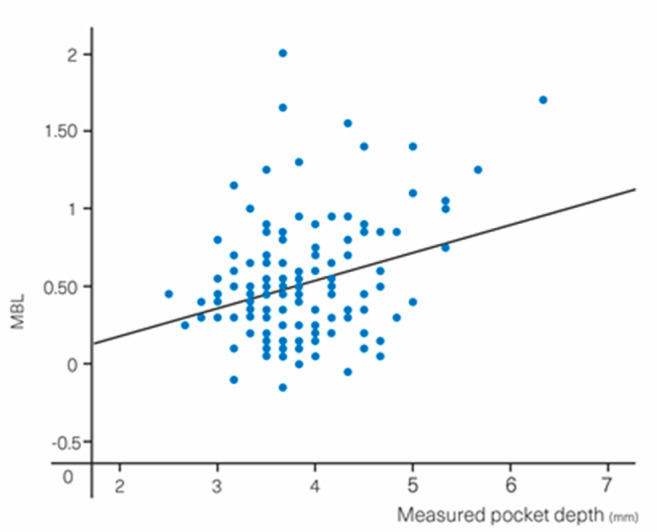
MBL vs. measured pocket depth.

**Figure 6 materials-13-04511-f006:**
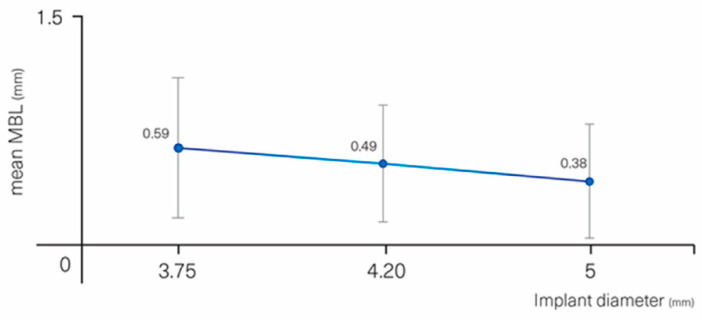
MBL vs. implant diameter.

**Table 1 materials-13-04511-t001:** Variables recorded in this study.

Patient Variables	Implant Variables	Surgical Variables	Prosthetic Variables
Age	Diameter	Corono-apical implant placement depth	Screw-retained
Gender	Length	Bone/sinus grafting	Cemented
Smoking	Local site	Healing protocol	Crown–implant ratio
Periodontal disease	Jaw	Insertion torque	-
Diabetes	Abutment height	-	-
Oral hygiene	Soft tissue thickness	-	-
Bone quality	Phenotype	-	-
-	Probing depth	-	-
-	Keratinized mucosa	-	-
-	Bleeding on probing	-	-

**Table 2 materials-13-04511-t002:** Probability of failure according to independent variables: Wald chi [2] test results of the simple binary logistic regression model. No odds risk (OR) could be calculated, because no failure occurred. No variable contributed to failure in a statistically significant manner; only one variable showed a tendency (*p* < 0.143).

Implant Failure	Category	OR	IC 95%	*p*-Value
SEX	Male (n = 24)	1	-	-
	Female (n = 43)	0.97	0.15–6.39	0.972
SMOKING	No (n = 59)	1	-	-
	Yes (n = 8)	1.86	0.16–21.2	0.617
DIABETES	No (n = 53)	1	-	-
	Yes (n = 4)	2.64	0.23–29.9	0.432
HISTORY OF PERIODONTITIS	No (n = 51)	1	-	-
	Yes (n = 26)	3.05	0.30–31.3	0.348
SEGMENT	Anterior (n = 40)	1	-	-
	Posterior (n = 90)	0.25	0.04–1.60	**0.143**
ARCH	Maxilla (n = 63)	1	-	-
	Mandible (n = 67)	1.22	0.20–7.47	0.832
DIAMETER (mm)	≤3.75 (n = 47)	1	-	0.981
	4.00–4.30 (n = 69)	1.02	0.16–6.49	0.981
	5 (n = 14)	-	-	-
LENGTH (mm)	8,0 (n = 24)	1	-	0.099
	10 (n = 58)	-	-	-
	11.5 (n = 40)	6.94	0.69–69.5	0.099
	≥13 (n = 8)	-	-	-
SURGICAL PROTOCOL	1 stage (n = 50)	1	-	-
	2 stage (n = 80)	1.81	0.17–18.8	0.62

**Table 3 materials-13-04511-t003:** Association between total MBL and independent variables of patient profile, surgery, and implant characteristics: Wald chi [2] test results of the general linear regression model.

Independent Variables	Category	Beta	IC 95%	*p*-Value
SEX	Male (0.48 ± 0.38)	0		
	Female (0.54 ± 0.41)	0.06	−0.13–0.25	0.515
SMOKING	No (0.52 ± 0.40)	0		
	Yes (0.50 ± 0.35)	−0.02	−0.22–0.18	0.839
DIABETES	No (0.55 ± 0.40)	0		
	Yes (0.32 ± 0.27)	−0.23	−0.43–−0.02	**0.035 ***
PERIODONTITIS	No (0.51 ± 0.32)	0		
	Yes (0.54 ± 0.46)	0.03	−0.16–0.22	0.753
SEGMENT	Anterior (0.48 ± 0.33)	0		
	Posterior (0.45 ± 0.42)	0.06	−0.08–0.20	0.391
ARCH	Maxilla (0.60 ± 0.46)	0		
	Mandible (0.46 ± 0.33)	−0.14	−0.29–0.02	0.078
DIAMETER (mm)	≤3.75 (0.64 ±0.19)	0		0.221
	4.00–4.30 (0.69 ± 0.21)	−0.10	−0.30–0.10	0.315
	5 (0.38 ± 0.36)	−0.22	−0.47–0.03	0.083
LENGTH (mm)	8 (0.58 ± 0.43)	0		0.749
	10 (0.50 ± 0.39)	−0.08	−0.24–0.08	0.338
	11.5 (0.53 ±0.41)	−0.05	−0.26–0.17	0.662
	≥13 (0.42 ± 0.27)	−0.12	−0.37–0.14	0.375
SURGICAL PROTOCOL	1 STAGE (0.56 ± 0.44)	0		
	2 STAGE (0.50 ± 0.37)	−0.06	−0.27–0.15	0.555
IMPLANT SITE	Healed (0.52 ± 0.40)	0		
	Post-extraction (0.48 ± 0.34)	−0.04	−0.29–0.21	0.728
BONE GRAFTING	No (0.51 ± 0.35)	0		
	Yes (0.58 ± 0.56)	0.08	−0.21–0.36	0.591
SINUS GRAFTING	No (0.51 ± 0.37)	0		
	Yes (0.66 ± 0.61)	0.15	−0.27–0.57	0.478
DEPTH OF IMPLANT PLACEMENT (mm)	≤−1.5 (0.86 ± 0.61)	0		**0.002 ****
	−1 mm (0.58 ± 0.35	−0.29	−0.71–0.14	0.189
	−0.5 mm (0.52 ± 0.35)	−0.34	−0.75–0.07	0.102
	≥0 mm (0.31 ± 0.29)	−0.55	−0.97–−0.13	**0.011 ***
SOFT-TISSUE PHENOTYPE	Thin (0.57 ± 0.33)	0		
	Thick (0.51 ± 0.41)	−0.06	−0.23–0.11	0.462
BONE QUALITY	1 (0.48 ± 0.54)	0		0.542
	2 (0.48 ± 0.32)	0	−0.34–0.34	0.983
	3 (0.60 ± 0.47)	0.12	−0.26–0.51	0.532
	4 (0.40 ± 0.26)	−0.08	−0.45–0.30	0.684

* *p* < 0.05; ** *p* < 0.01; *** *p* < 0.001.

**Table 4 materials-13-04511-t004:** Association between MBL and other clinical parameters: Wald chi [2] test results of the general linear regression model.

Parameter	Category	Beta	IC 95%	*p*-Value
BOP BUCCAL	No (0.48 ± 0.39)	0	-	-
	Yes (0.65 ± 0.40)	0.17	0.00–0.34	**0.049 ***
BOP LINGUAL	No (0.50 ± 0.38)	0	-	-
	Yes (0.63 ± 0.46)	0.13	−0.07–0.32	0.202
PD TOTAL	(0.52 ± 0.39)	0.18	0.05–0.31	**0.008 ****
PLAQUE BUCCAL	No (0.54 ± 0.41)	0	-	-
	Yes (0.46 ± 0.32)	−0.08	−0.20–0.06	0.259
PLAQUE LINGUAL	No (0.52 ± 0.41)	0	-	-
	Yes (0.49 ± 0.32)	−0.04	−0.22–0.15	0.708
KT	(0.52 ± 0.39)	−0.10	−0.17–−0.03	**0.004 ****
KT groups	<2 mm (0.78 ± 0.40)	0	-	-
	≥2 mm (0.45 ± 0.36)	−0.34	−0.51–−0.16	**<0.001 *****
Ti-BASE	No (0.43 ± 0.43)	0	-	-
	Yes (0.56 ± 0.37)	0.13	−0.09–0.35	0.235
MULTI-UNIT	No (0.55 ± 0.37)	0	-	-
	Yes (0.43 ± 0.44)	−0.12	−0.35–0.11	0.304
SOFT TISSUE PHENOTYPE	Thin (0.57 ± 0.33)	0	-	-
	Thick (0.51 ± 0.41)	−0.06	−0.23–0.11	0.462
GINGIVAL THICKNESS	-	−0.07	−0.22–0.08	0.384
BONE QUALITY	-	0.06	−0.09–0.21	0.416

* *p* < 0.05; ** *p* < 0.01; *** *p* < 0.001.

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
