# Peer review of "Factors Affecting Implant Failure and Marginal Bone Loss of Implants Placed by Post-Graduate Students: A 1-Year Prospective Cohort Study"

_materials, 2020, doi:10.3390/ma13204511_

Round 1

Reviewer 1 Report

The topic is somehow relevant, especially if the person wants to be treated by a post-graduate student or an unexperienced surgeon. It is also useful for the evaluation of the teaching methods used so far and lead to potential improvements. It seams that there isn't so much studies about the subject and that means the novelty of the study. Anyway regarding the success of post graduate students some comparison is missing with a control population of experienced surgeons. This comparison is proposed in literature references (line 75) and (line 259/260) about this study. In the conclusions, 1, the the 3.6% failure rate is considered low, in comparison to what? This is where a control population is missing. This study would benefit from a more thorough comparative analysis with the work performed by experienced surgeons in the same environments. Even with that consideration that doesn't take the relevance of the study in terms of evaluation of the training effectiveness for post graduate students in this area.

Author Response

Thank you very much for your positive comments.

In the conclusion we considered 3.6% of failures as a rather low failure rate. This assertion is based on the results published in the general literature dealing with implant failure, as explained in the discussion. It relies also more specifically on the paper of Horwitz et al. (2018) that involved 7 highly experienced practitioners that placed and rehabilitated the very same C1 implants and came to similar success and survival rates at the 1-year follow-up.

Obviously, the reviewer is right to mention that this study would have been strengthened with an internal control group of experienced practitioners. However, organization and protocols of the University make that all patients are treated exclusively by students; a control group with experienced teams is out the possibilities of such a study in such an environment. In addition, what is considered as an experienced surgeon is open to discussion as it appears in the Introduction §. 

Bottom line of this paper is that very few studies have explicitly addressed the issue of implants placed by inexperienced students and this study contributes humbly to add to the literature. Readers that might have thought that attending a University setting with students must be avoided at all costs might have with this paper second thoughts as far as the students are supervised.

Reviewer 2 Report

Thank you for the interesting research.

Brush up the abstract, for example, in Line 25 UIC will not come in later, so no need to use UIC, just write international university of Catalonia.

And in line 34, C1 MIS is not common word, just write dental implant and another C1 implants in abstract should be changed to dental implants or implants.

In introduction at Line 78, Company name and country and so on for C1 implant should be added.

Line 84

This study is prospective study, written in title, and all patients who came to the university hospital were included initially or basically some patients were selected from all cohort? Some patients received another type of implant and they were rejected in this study you mean? or  only you analysed the patients with C1 implant and that is the retrospective procedure?

There are a lot of variables included in this study, and it should be smaller numbers of variables. That is too complicated.

Line 148-157

Which type of probe did you use?

Author Response

Thank you very much for your positive comments. 

We have made the suggested changes to the manuscript in the Abstract and the Introduction. 

The study has a prospective design: we included all the patients who attended the Hospital between January 2016 and January 2017; and all patients received C1 implants. We have analyzed them in a cohort study to be able to evaluate the outcome of these implants in many clinical situations.

We have included many variables to make a single paper that has more power compared to making several articles with fewer variables. We agree that a lot of information have been collected and exposed here but we think that it makes this paper even a more relevant and informative reading.

We used a periodontal probe (UNC 15, Hu-Friedy) as reported in material and methods

Reviewer 3 Report

The manuscript (materials-953886) entitled “Factors Affecting Implant Failure and Marginal Bone Loss of Implants Placed by Post-Graduate Students: a  1-Year Prospective Cohort Study” presents the 1-year results evaluating possibility of dental implant failure after surgical procedures performed by post-graduate students.

This manuscript is interesting but some major concerns should be taken into account:

Abstract:

1) Why did the authors choose C1, MIS implants? It should be clarified

2) The short description about C1, MIS implants should be introduced to the text.

Introduction:

1) The description about C1, MIS implants should be introduced to the text.

Materials and Methods:

2.1 Study design

Why did the authors evaluate the success of implantation only during 1-year? Maybe, this time is too low to obtain reliable results about implant failure. In my opinion, the research should be perform at least by 3 years.

2.3 Surgical procedure

The authors wrote: “All the surgical and prosthetic procedures were performed at the Clinic of Dentistry of the 110 University by 24 post-graduate students”. It is important to know whether students are the same age? It is important to determine whether they were graduated at the same time. I am scientist and academic teacher, so sometimes I see a big differences between the level of student skills, whose are in the same student’ group. Likewise, the students graduated at different years possess various skills. Thus, were the authors sure that 24  students may be a reliable group?  

Moreover, as authors mentioned that the post-graduate students were under the care of experienced doctor, so the possibility to make mistakes significantly decrease. If even, the students may make mistake lead to failure of implant, they were most likely identified and corrected.  

The another important point is associated with the place where the students graduated. I suppose that it was Universitat Internacional de Catalunya. So, in my opinion this study cannot provide the general conclusion, because it only evaluate the skills of post-graduate students from this University. So, it will be great to expand the area of research (I mean co-operation with others Clinics of Dentistry or others University).

Results:

3.1 Implant survival

What was a reason that 5 implants failed before loading?

Author Response

Thank you very much for your encouraging comments. 

We have made the suggested changes to the manuscript and we introduced a brief description in the Material and Methods section. We have used the C1 implant because this study was part of a grant funded by MIS and these implants were available at the University.

In this report, we evaluated the results at the 1-year follow-up in order to share the short-term outcome of this study that provides information that are scarcely found in the literature, i.e. implant therapy realized by unexperienced students. It is common practice in the literature to publish 1-year data because this timeline, although short, provides a first appreciation of the clinical situation. Nowadays with implants that have been in use for a certain time (here for 10 years) and following accepted protocols, it is exceptional that substantial differences are seen between the 1-year and 3-year follow-up. But the reviewer is right, and this is the reason why we are presently reviewing the patients for the 3-year data collection that will be published in a later stage.

Students belonged to the 1st and 2nd specialization years. All just graduated the Dental School and were 24 to 27 years old; this was added in the corresponding §. They were not colleagues with a private practice experience that attended implant specialization. We think therefore that is it a homogeneous group with obviously disparate skills as in all groups, but all share in common that they had inexperience hands and were supervised by an assistant as it is common practice in an academic setting. In addition, a group of 24 distinct students is large enough to have a large variety of talents and skills. 

We understand your concern about the place from where the students graduated; all were from the local Universidad Internacional de Catalunya. Expanding the area of research with others University would have given more strength to the study but, as a scientist and academic teacher, you most probably know how complex it is to organize a prospective multi-center study involving distinct Universities. This research was made possible to conduct because our University received a grant from MIS. 

The point you are raising about the local quality of the University teaching sounds to be based on the reality of your long experience. But what you mentions applies as well to all papers published by famous clinicians. And we don't see in review papers the mention that the data under scrutiny have been published by skilled and renowned clinicians and all other should be cautious when they are using it and expect dramatically distinct results.

We didn’t find a parameter to significantly affect implant failure. Only a relevant association between survival and implant position (anterior vs. posterior) was found in which the failure risk was 4 times lower in posterior implants than in anterior ones but did not reached statistical significance.

No specific reason for failure could be established with certainty because of the low number of failures. However, we mentioned that more failures occurred in perio involved patients although not in statistically significant way.

Round 2

Reviewer 2 Report

Thank you very much for the modification following my suggestions.

My parts are now acceptable.

Congratulations.

Reviewer 3 Report

I accepted the authors response, but I recommend to perform the similar research including longer time and more places (i.e., other University).